# The Role of TNF-α in Alzheimer’s Disease: A Narrative Review

**DOI:** 10.3390/cells13010054

**Published:** 2023-12-26

**Authors:** Domenico Plantone, Matteo Pardini, Delia Righi, Carlo Manco, Barbara Maria Colombo, Nicola De Stefano

**Affiliations:** 1Department of Medicine, Surgery and Neuroscience, University of Siena, Viale Bracci 2, 53100 Siena, Italy; d.righi4@student.unisi.it (D.R.); carlooo95.cm@gmail.com (C.M.); destefano@unisi.it (N.D.S.); 2Department of Neuroscience, Rehabilitation, Ophthalmology, Genetics, Maternal and Child Health, University of Genoa, L.go P. Daneo 3, 16132 Genova, Italy; matteo.pardini@gmail.com; 3IRCCS Ospedale Policlinico San Martino, Largo Rosanna Benzi 10, 16132 Genova, Italy; barbaramaria.colombo@hsanmartino.it

**Keywords:** Alzheimer’s disease, dementia, inflammation, cognitive impairment, TNF-α

## Abstract

This review analyzes the role of TNF-α and its increase in biological fluids in mild cognitive impairment, and Alzheimer’s disease (AD). The potential inhibition of TNF-α with pharmacological strategies paves the way for preventing AD and improving cognitive function in people at risk for dementia. We conducted a narrative review to characterize the evidence in relation to the involvement of TNF-α in AD and its possible therapeutic inhibition. Several studies report that patients with RA and systemic inflammatory diseases treated with TNF-α blocking agents reduce the probability of emerging dementia compared with the general population. Animal model studies also showed interesting results and are discussed. An increasing amount of basic scientific data and clinical studies underscore the importance of inflammatory processes and subsequent glial activation in the pathogenesis of AD. TNF-α targeted therapy is a biologically plausible approach for cognition preservation and further trials are necessary to investigate the potential benefits of therapy in populations at risk of developing AD.

## 1. Introduction

Alzheimer’s disease (AD) is a progressive neurodegenerative disorder that represents the most common cause of dementia in the elderly [1]. Typical clinical manifestations include declarative memory deterioration, temporal and spatial disorientation, language difficulties, judgment problems, and other cognitive impairments [2]. The key pathological feature of AD is the accumulation of beta-amyloid (Aβ) plaques and neurofibrillary tangles [3]. Aβ deposits supposedly disrupt normal synaptic communication and, in the end, may contribute to neurodegeneration [4]. However, this “classical” pathogenic model does not explain much of the clinical and pathological variability of the disease; therefore, several additional hypotheses have been formulated [5]. The “inflammatory hypothesis” of AD [6,7] suggests that chronic inflammation in the brain plays a significant role in the development and progression of the condition. Ongoing immune system activation and the release of pro-inflammatory molecules contribute to the characteristic AD pathological changes, with the final result of damaging neurons and impairing the clearance of toxic substances [8,9,10,11]. The intricate interplay between the central nervous system (CNS) and the peripheral immune system has come into focus [12,13] revealing altered levels of several cytokines in the peripheral blood and cerebrospinal fluid (CSF) of patients with AD [14,15,16,17]. Tumor necrosis factor (TNF)-α, a proinflammatory cytokine, has attracted attention due to its multifaceted and still not fully characterized role in AD and the fact that it could potentially represent a future therapeutic target. Indeed, TNF-α has many roles, not only in inflammation, but also in cell death and proliferation, and is crucial in maintaining CNS homeostasis [18,19,20]. TNF-α is primarily produced by microglia, astrocytes, and neurons in response to various stimuli [21]. Although TNF-α has a fundamental role in modulating excitotoxicity, neuroinflammation, blood-brain barrier permeability [19], regulation of oligodendrocyte survival, myelin formation and repair [22], its excessive or dysregulated production can have detrimental effects on neuronal health, contributing to the development of neurodegeneration [23].

In this narrative review, we explore the role of TNF-α in the pathogenesis of AD. There are multiple pieces of evidence regarding the role of TNF-α, including its ability to induce amyloid precursor protein (APP) and β-site APP cleaving enzyme (BACE1) expression in mouse astrocytes [24,25,26], its stimulation of Aβ synthesis, neuronal loss, and impairment microglia Aβ phagocytosis [27].

We first discuss the molecular structure, receptors, and metabolism of TNF-α. We explore TNF-α molecular structure, with reference to its membrane-bound form and its soluble form, the two TNF-α receptors, and the metabolism of this cytokine. Then, we explain the rationale for the TNF-α involvement in AD. In this part, we summarize the activities in which TNF-α is involved within the CNS, the roles of TNF-α receptors in CNS physiology, and how TNF-α influences and curbs neuroinflammation in AD. We also delved into both preclinical and clinical studies providing valuable cues for the role of TNF-α in AD. Therefore, we summarize the evidence in humans, also discussing clinical trials exploring the efficacy of anti-TNF-α drugs on the progression of AD and the main evidence coming from animal studies in AD models. We ultimately deal with future directions, with reference to the possible applications of TNF-α inhibition to slow down the progression of cognitive function and possibly even prevent AD onset [28,29,30].

## 2. Materials and Methods

To select the relevant literature for this narrative review, the authors first searched the PubMed, Embase, and Google Scholar databases as well as their personal article libraries using the following string: TNF-α AND Alzheimer’s Disease; serum TNF-α AND Alzheimer’s Disease, CSF TNF-α AND Alzheimer’s Disease; neuroinflammation AND Alzheimer’s Disease. Articles not in the English language, predating the 1980s, or existing solely in abstract form were excluded unless deemed pertinent. Subsequently, the authors assessed the abstracts of all retained articles, specifically selecting those aligned with the objectives of the current narrative review. The authors then conducted a rigorous examination of the selected articles, excluding those of subpar quality based on criteria such as the number of enrolled subjects, clinical relevance of presented data, methodological robustness, and the presence of redundant information.

The initial research included 198 articles. After careful evaluation, 115 of them were used for the present review, of which 4 were the main studies exploring the efficacy of anti-TNF-α drugs on the progression of AD and 14 were the main selected papers on animal studies.

## 3. Molecular Structure, Receptors and Metabolism

### 3.1. Molecular Structure

The term “tumor necrosis factor” was first coined by O’Malley and colleagues in 1962 and this molecule was initially recognized for its role in initiating apoptosis [31]. The TNF-α gene is situated in the class III region of the major histocompatibility complex on chromosome 6, found between the HLA-B and HLA-DR genes [32]. TNF is present in two distinct forms: a membrane-bound form and a soluble form [33]. In the process of its synthesis, TNF-α initially stays within the double layer of the cell membrane. Subsequently, a specific enzyme, known as the TNF-converting enzyme (TACE/ADAM17/CD156q) [33], liberates the membrane-bound TNF-α, which is originally 26 kDa in size (233 amino acid residues), into the extracellular environment, where it becomes a soluble 17 kDa protein, through a phenomenon called “shedding” [34]. Membrane-bound TNF-α is cleaved by TACE between residues of alanine^76^ and valine^77^ [35]. Soluble TNF-α forms a homotrimer comprising 17-kDa cleaved monomers, while transmembrane TNF-α also exists as a homotrimer with 26-kDa uncleaved monomers [36]. Membrane-bound and soluble forms of TNF-α are both biologically active, but their specific functions vary based on their respective locations [37].

### 3.2. TNF Receptor 1 (TNFR1) and TNF Receptor 2 (TNFR2)

There are two receptors for TNF-α, named TNF receptor 1 (TNFR1) and TNF receptor 2 (TNFR2). Membrane-bound TNF-α exhibits a primary interaction with TNFR2 but can also bind TNFR1 [38]. Soluble TNF-α functions in an endocrine and paracrine manner, as it can travel through the bloodstream to distant sites [39].

Soluble TNF-α functions are mediated through TNFR1 and TNFR2 and, interestingly, it has been shown that TNFR1-mediated signaling pathways predominantly occur via the soluble form of TNFα. Additionally, this soluble form has the capacity to initiate the clustering of TNFR1 on the plasma membrane of living cells [40].

Both types of TNF receptors initiate unique and shared signaling pathways that can function in a coordinated manner. Additionally, alternative splicing or shedding processes have been documented to generate soluble forms of TNFR1 and TNFR2. These soluble variants of TNF receptors hinder TNF activity by competing for TNF binding with cellular receptor species and may also function as dominant-negative molecules [41,42,43]. TNFR1 serves as a death receptor and possesses a “death domain” within its cytoplasmic segment [44]. The “death domain” represents a conserved protein–protein interaction domain, facilitating homotypic interactions between death receptors and cytoplasmic proteins that also contain a “death domain” [45]. Signaling proteins containing “death domains” connect TNFR1 to cytotoxic pathways, triggering apoptosis or necroptosis. Additionally, they facilitate the activation of signaling pathways that involve transcription factors from the nuclear factor of kappa B (NFκB) family or kinases from the MAP kinase family [46,47,48].

On the contrary, TNFR2 lacks a death domain and represents a prototype TNF receptor-associated factor (TRAF)-interacting member within the TNFR superfamily [49]. TRAF family proteins are centrally involved in modulating inflammation, orchestrating antiviral defenses, and regulating apoptosis [49]. Consequently, a brief amino acid motif near the C-terminus of TNFR2 facilitates the recruitment of the adapter protein TRAF2 and associated proteins, including TRAF1, cellular inhibitor of apoptosis protein 1 (cIAP1), and cIAP2 [49]. TNFR2, therefore, does not possess inherent cell death-inducing activity but instead stimulates NFκB signaling and the activation of various kinases [48]. It has been shown that TNFR2 activation induces a long-lasting NF-κB activation, which is independent of the TNFR1 signaling pathway [50]. TNFR2 also activates the phosphoinositide 3-kinase (PI3K)-protein kinase B (Akt) pathway, contributing to various cellular functions [50]. The PI3K-Akt pathway serves as an intracellular signaling mechanism, driving essential cellular processes such as metabolism, proliferation, cell survival, growth, and angiogenesis in response to external signals. This regulatory cascade involves the phosphorylation of downstream substrates, particularly on serine and/or threonine residues [50].

### 3.3. TNF-α Metabolism

TNF-α has a relatively short half-life once released into the extracellular environment, typically ranging from 20 min to a few hours [51]. This rapid turnover enables TNF-α to respond promptly to alterations in the microenvironment. Following the release, TNF-α undergoes liver and kidney metabolism and clearance [52].

## 4. Rationale for the TNF-α Involvement in AD

Increasing scientific evidence consistently confirms the significance of immunity in the pathogenesis of AD. These findings increasingly suggest a combined involvement of both innate and adaptive immunity. TNF exhibits a continual, low-level expression in the typical adult brain [53,54,55,56]. TNF-α is secreted by many cell types, albeit cells of the monocytic lineage, including microglia and are the primary synthesizers [57]. TNF-α modulates and influences many neuronal activities in the CNS, which include glutamatergic transmission [58,59,60] and gamma-aminobutyric acid (GABA) transmission [58,61,62], with an overall outcome of these alterations of an augmentation of excitatory synaptic transmission compared to inhibitory transmission in hippocampal slices [58] and potentially in other neuronal circuits [56]. The strengthening of excitatory transmission is further amplified by the release of glutamate from astrocytes influenced by TNF-α [56,63].

TNF-α is involved in a variety of activities within the CNS. It influences sleep [64] by activating NF-κB pathways, leading to the upregulation of adenosine A1 receptors [65]. TNF-α directly acts on neurons in key sleep regulatory areas of the brain, such as the hypothalamic preoptic area and basal forebrain, impacting sleep architecture [66]. Intracerebroventricular injection of TNF-α induces a significant increase in slow-wave sleep, as demonstrated by polysomnography, emphasizing its role in promoting specific sleep stages [67]. In mice, the administration of TNF-α to the peritoneum leads to significant elevations in non-rapid eye movement sleep duration while causing reductions in slow wave activity [68].

The roles of TNF-α receptors in CNS physiology are different. TNFR1 exhibits neuroprotective effects in the CNS by safeguarding cells against necrosis [69]. Despite the anti-apoptotic attributes of TNFR1 signaling, it only partially elucidates its neuroprotective role in acute CNS injuries. Notably, TNFR1 fails to shield neurons from insults like retinal ischemia–reperfusion injury and oligodendrocytes from immune-mediated damage in conditions such as multiple sclerosis (MS) and experimental autoimmune encephalomyelitis (EAE) [56]. Studies exploring caspase inhibition as a potential remedy for TNF toxicity yielded surprising results. Pan-caspase inhibition exacerbated TNF toxicity in mice, intensifying oxidative stress and mitochondrial damage [70]. TNFR1-mediated cell death involves a switch between caspase-induced apoptosis and necrosis, with key signaling molecules such as receptor-interacting protein 1 (RIP1) and RIP3 identified in TNF-induced programmed necrosis or necroptosis [71,72]. This reveals TNFR1 signaling and the Fas-associated death domain protein (FADD)/caspase 8 apoptotic platform as an additional protective layer, enabling cells to undergo apoptosis instead of inflammation-inducing necroptotic cell death. The neuroprotection mediated by TNFR1 in the CNS may operate on two levels: gene induction and de novo production of survival molecules and, in the absence of adequate survival signals, caspase 8-dependent apoptosis [56].

TNFR2 also plays a crucial role in neuroprotection and repair processes in the CNS. TNFR2 is expressed in regulatory T cells, endothelial cells [73], oligodendrocyte lineage cells [74], and specific neuron populations [50] and is able to activate pro-survival signaling through TRAF2 recruitment [75] and PI3K/NF-κB pathway activation [76,77]. TNFR2 activation is associated with Akt/protein kinase B activation, crucial for neuroprotection against excitotoxicity and ischemia–reperfusion injury [56]. Actually, TNFR2 and TNFR1 cooperate to enhance neuroprotection, with additive effects on cell survival through NF-κB activation. Indeed, TNFR2 reinforces TNFR1-mediated cell survival and apoptotic signaling, maximizing neuroprotection [77,78]. Finally, TNFR2 may also promote oligodendrocyte precursor cell proliferation and remyelination, contributing to repair mechanisms [78,79].

However, TNF-α represents one of the pivotal pro-inflammatory cytokines and triggers the production of IL-1, IL-6, and IL-8, fostering chronic inflammation unless balanced by anti-inflammatory cytokines like IL-10 [80]. It induces APP and BACE1 expression in mouse astrocytes, activating γ-secretase in HEK cells, and releasing Aβ peptides [24,25,26]. Persistent brain inflammation forms a self-amplifying cycle, sustaining high TNF-α levels, potentially stimulating Aβ synthesis, causing neuronal loss, and hindering microglia Aβ phagocytosis [27]. The role of TNF-α in tau hyperphosphorylation is less understood, but few interesting data on mice suggest a possible connection [81,82].

Historically, microglia activation and gliosis were perceived as secondary to neurodegeneration in AD. However, recent genetic investigations in late-onset AD indicate the involvement of microglia- and astrocyte-related pathways together with the activation of multiple immune pathways in the very early stages of the disease [6,10,11,83,84,85,86,87,88]. Particularly, the innate immune receptor triggering receptor expressed on myeloid cells 2 (TREM2), expressed on microglia and myeloid cells, has garnered attention [89,90]. Rare TREM2 mutations linked to AD propose that TREM2 deficiency plays a role in AD susceptibility [91]. Functioning as a negative regulator, TREM2 modulates the release of TNF-α and other inflammatory cytokines via the Toll-receptor pathway [92]. Impairment of TREM2 function in monocytes or macrophages may contribute to systemic TNF-α production, potentially serving as a treatable risk factor for AD [91,93].

The TNF-α gene promoter region contains multiple single nucleotide polymorphisms (SNPs), with the G308A mutation being of interest for AD [94]. This mutation elevates TNF-α mRNA and protein expression and a possible pathogenetic mechanism has been proposed. However, conflicting meta-analyses reveal regional variations, emphasizing the need for further investigation into the correlation between TNF-α G308A and AD risk [95].

Altogether, these insights underscore the intricate interplay of immune responses in AD pathogenesis and highlight the role of TNF-α. In the following paragraph, we will review the main evidence in human and animal studies and finally, we will discuss the possible future directions.

## 5. Evidence in Humans

### 5.1. Cerebrospinal Fluid and Blood Studies

The primary evidence for this association derives from an initial study performed on the CSF of mild cognitive impairment (MCI) patients which demonstrated higher TNFα and tau protein levels and lower TGFß and Aß levels in MCI patients compared to controls. Interestingly, MCI patients progressing to AD showed elevated CSF TNFα [96]. Elevated serum TNF-α concentrations in AD patients compared to healthy individuals and individuals with MCI were subsequently demonstrated [15,17,97]. A 6-month study exploring the association between TNF-α levels and cognitive performance in 300 subjects with varying AD severity, found that acute inflammation correlated with a twofold cognitive decline increase, and high-baseline TNF-α levels quadrupled the decline [98]. Kim and colleagues performed an analysis of serum cytokine levels in individuals affected by AD, in those with MCI, and in a cohort of healthy controls to ascertain the connections between these cytokine levels and the neuropsychological parameters. Noteworthy associations between the levels of TNF-α and interleukin-6 (IL-6) and the cognitive performance, assessed using the Mini-Mental State Examination (MMSE) score were described [16]. Another interesting study explored the correlation between CSF TNF-α and functional connectivity in 64 older adults and found that elevated TNF-α levels correlated with reduced connectivity in decision-making, inhibitory control, and memory regions, with APOE4 status moderating this effect [99]. The time courses of levels of multiple plasma and CSF cytokines in patients with AD and age-matched control subjects were assessed in a study by Llano and colleagues, by measuring cytokine levels 7 times over 24 h in plasma and CSF using a lumbar catheter. The authors found that CSF levels of IL-1β, IL-2, IL-10, IL-12p70, granulocyte-macrophage colony-stimulating factor, interferon-γ, and TNF-α diverged over time, with higher levels in AD subjects compared to controls, with no difference in cytokine trajectories observed in plasma [100]. Another noteworthy investigation revealed that TNFR displayed more robust connections with total tau and p-tau in comparison to Aβ1-42, spanning healthy controls, MCI, and AD subjects. Within the longitudinal cohort, individuals with MCI who exhibited heightened levels of CSF TNFR1 and diminished levels of TNFR2 had an increased susceptibility to advancing to AD [101].

### 5.2. Genetic Studies

Moreover, extensive research has suggested that specific TNF-α gene polymorphisms contribute to an increased risk of AD. In a meta-analysis aiming to define the association of common TNF-α gene polymorphisms with the risk of AD, Di Bona et al. selected 17 studies and evaluated them with a model-free method approach, to comprehensively analyze the results of these case-control genetic association studies. Notably, their research indicates a correlation between the −850 C > T polymorphism and the susceptibility to AD [102]. Additionally, Yang et al. performed a case-control study in the Southern Chinese population, involving the use of polymerase chain reaction-sequence specific primers (PCR-SSP) to assess TNF-α genotypes and alleles in 112 sporadic AD patients and 121 controls. Additionally, they quantified serum TNF-alpha levels through radioimmunoassay. They found significantly higher levels of serum TNF-α in sporadic AD patients and that both −308 A/G polymorphism and elevation of serum level of TNF-α were both associated with an increased risk of AD [103].

### 5.3. TNF-α Inhibitors

Observational investigations conducted on individuals with systemic inflammatory disorders treated with TNF-α inhibitors offer additional substantiation for the implication of TNF-α in the pathogenesis of AD. A large retrospective case-control analysis, encompassing 56 million adult patients afflicted with inflammatory conditions, revealed a diminished risk of AD development with the administration of TNF-α inhibitors [93]. A previous nested case-control study revealed a higher AD prevalence among RA patients compared to those without rheumatoid arthritis, with a further significant increase in risk in those who were affected by chronic conditions including coronary artery disease, diabetes, and peripheral vascular disease. Notably, exposure to anti-TNF agents, particularly etanercept, was associated with a reduced risk of AD in these patients [104]. Elderly rheumatoid arthritis patients showed improved cognitive performance with subcutaneous anti-TNF-α therapy using drugs like etanercept and adalimumab [105]. In contrast, a double-blind study in mild and moderate AD patients treated with subcutaneous etanercept did not show significant changes in cognitive function, behavior, and global functions though there was a positive trend in the anti-TNF-α treatment group [106]. The limited effectiveness of subcutaneous anti-TNF-α therapies in AD patients may be attributed to the large molecular weight of anti-TNF-α monoclonal antibodies because this makes the passage through the blood–brain barrier impossible under physiological conditions [82].

Interesting clinical studies collectively suggested that perispinal administration of etanercept may lead to significant and sustained improvements in AD and primary progressive aphasia, showcasing its potential as a therapeutic intervention. In a prospective, single-center, open-label, pilot study, involving 15 patients with AD and primary progressive aphasia, perispinal etanercept infusion (25–50 mg weekly for six months) demonstrated significant improvement in cognition [107]. The benefits of this treatment were sustained in patients who continued for over two years [108]. Furthermore, another case report describing an 81-year-old patient who was given 25 mg of perispinal etanercept by posterior cervical interspinous injection outlined swift cognitive enhancement, starting within minutes. The rapid cognitive improvement observed in this patient has been hypothesized to be linked to the mitigation of the effects of excess TNF-α on gliotransmission or other synaptic mechanisms in AD [109,110]. Due to the limitations of large molecules like etanercept in crossing the blood–brain barrier with traditional systemic administration, the perispinal direct drug delivery system likely played a role in the treatment’s efficacy [110].

Studies exploring the efficacy of anti-TNF-α drugs on the progression of AD are summarized in Table 1. To the best of our knowledge, there is currently no ongoing study that can adequately elucidate the potential role of these drugs in AD. Further research and larger-scale studies are warranted to establish the efficacy of TNF-α inhibition in AD conclusively.

## 6. Evidence in Animal Models of Alzheimer’s Disease

Since the beginning of the 2000s, researchers have conducted tests on animal models of AD using TNF-α inhibitors as potential therapeutic drugs targeting disease progression. The tested TNF-α inhibitors include minocycline, infliximab, imipramine, hydrogen sulfide (NaHS), thalidomide, 3,6-Dithiothalidomide (3,6-DT), exendin 4, etanercept and TfRMAb-TNFR, the latter being a biologic TNF-α inhibitor with the capability to penetrate the blood–brain barrier, effectively sequestering TNF-α in both peripheral blood and CNS regions.

This diverse range of compounds, evaluated in animal models, showcases the multifaceted approaches undertaken in the pursuit of finding effective interventions for AD by targeting the inflammatory pathways associated with TNF-α. In the upcoming sections, we will review and analyze key studies on animal models that contribute to our understanding of the therapeutic potential of TNF-α inhibition for AD.

### 6.1. Minocycline

One of the anti-TNF-α drugs was minocycline, which is a tetracycline derivative with anti-inflammatory properties that can cross the blood–brain barrier and decrease the TNF-α production in the brain [111,112,113]. Researchers, led by Seabrook, explored minocycline potential as an anti-TNF-α agent and its interactions with microglia and astrocytes, typically activated in the presence of Aβ plaques [114]. Minocycline was administered to J20 APP-tg mice for three months, either in their youth or after Aβ deposition in 5–7 months. Results revealed distinct outcomes in relation to the timing of minocycline administration. When given to younger mice, there was a marginal increase in Aβ deposition in the hippocampus, yet cognitive performance improved. Conversely, administering minocycline post-Aβ deposition suppressed microglial activation without impacting Aβ levels or cognitive function [114].

Another study on minocycline was performed by Biscaro and her colleagues, which focused on the role of microglia in neurogenesis in the presence of Aβ. Microglia, in fact, has a dual effect, on one hand, can have both beneficial on the other hand, can have detrimental effects. It can be inhibited by the administration of minocycline. They injected minocycline in APP/PS1 mice that were 11 weeks old, showing that minocycline treatment increased the survival of dentate granule cells in the AD-mimicking mice, thanks to immunohistochemical stain testing. In this study, the authors demonstrated that Aβ levels within the brain are not influenced by minocycline. On the other hand, a possible promotion of hippocampal neurogenesis has been described for minocycline. This effect may be very interesting as a future therapeutic pathway to explore [115].

Overall, minocycline’s timing is crucial: it improves cognition in youth and suppresses microglial activation post-Aβ accumulation.

### 6.2. Imipramine

Chavant et al. explored the impact of imipramine on mice by intracerebroventricularly injecting Aβ25-35 peptides, followed by imipramine treatment starting the day after injection [116]. Imipramine, a tricyclic antidepressant, exhibits potential immunomodulatory effects in AD by inhibiting TNF-α and Aβ deposition [117]. The study assessed memory function using the Morris water-maze test and Y-maze tasks ten days post-injection. Notably, imipramine demonstrated a capacity to reduce elevated Aβ levels in both the frontal cortex and hippocampus, influencing the modulation of anti-C-terminal fragments/APP [116]. In conclusion, imipramine, an antidepressant, reduces Aβ levels, showing potential immunomodulatory effects in AD.

### 6.3. Hydrogen Sulfide

In their study, Xuan et al. delved into the effects of NaHS in male Wistar rats exhibiting neuroinflammatory responses and cognitive impairment following exposure to Aβ1-40 in the hippocampus. Their findings revealed that NaHS effectively mitigates the presence of Aβ1-40 in the hippocampus, demonstrating a possible neuroprotective role. The cognitive improvement was assessed using the Morris water maze test, indicating a positive impact on learning abilities. Furthermore, NaHS treatment induced a reduction in the expression of TNF-α and IL-1β mRNA, as determined by RT-PCR analysis [118]. The collective results strongly suggest that pre-treatment with NaHS proves successful in enhancing learning and memory capabilities while concurrently diminishing neuroinflammation levels and implies a possible future therapeutical role for NaHS in neuroinflammatory conditions [119,120].

### 6.4. Thalidomide and 3,6-Dithiothalidomide

Thalidomide, a frequently utilized anti-TNF-α medication, attenuates TNF-α synthesis by elevating the rate of mRNA transcript degradation. Nevertheless, thalidomide is associated with significant adverse effects, including heightened production of oxygen radicals [121].

Gabbita and colleagues conducted a study examining the impact of thalidomide and its analog, 3,6-DT, administration. In 4-month-old 3xTg-AD mice, small inhibitors of TNF-α were injected, and cognitive performance was assessed using the radial arm maze task and behavioral procedure task after 10 weeks of treatment. The results revealed an enhancement in working memory performance. Subsequent to the treatment with 3,6-DT and thalidomide, immunohistochemical tests and RT-PCR evaluations were employed to assess the expression levels of TNF-α mRNA and its intracellular TNF-α protein. Notably, only 3,6-DT demonstrated efficacy in preventing cognitive impairment and reducing brain TNF-α gene expression levels. TNF-α mRNA expression was diminished following both treatments [122].

Belarbi and colleagues have reported that 3,6-DT exhibits greater potency than thalidomide itself [123]. The study aimed to evaluate the potential of 3,6-DT as a treatment for chronic neuroinflammation. F344 rats, aged three months, underwent lipopolysaccharide infusion to induce neuroinflammation. After a 28-day period, the rats received daily injections of 3,6-DT, which normalized TNF-α levels, although IL-1β remained elevated. Furthermore, 3,6-DT reduced the expression of genes associated with the Toll-like receptor signaling pathway. However, it did not impact the number of MHC Class II immune cells [123].

Another study evaluating the efficacy of 3,6-DT in 3xTg-AD mice was conducted by Tweedie and colleagues. The experimental protocol involved administering 3,6-DT to the mice, and after a 5-week period, the animals underwent spatial memory acquisition and retention testing using the Morris water maze. Additionally, the expression of TNF-α protein was assessed by RT-PCR and immunohistochemistry. The results demonstrated a notable reduction in AD hallmark features, including decreased levels of pTau protein, Aβ peptide, and Aβ plaque numbers. Furthermore, memory deficits, evaluated through Morris water maze testing, exhibited improvement following the 3,6-DT treatment [124]. In conclusion, in all three studies just seen, 3,6-DT treatment in AD mice reduces AD hallmarks—pTau, Aβ, and plaques—improving spatial memory and cognitive deficits.

### 6.5. Exendin-4

In 2012, Bomfim and colleagues directed their attention to the association between impaired brain insulin levels and cognitive impairments in AD patients [125]. Although previous studies had hinted at this correlation [126,127], their investigation aimed to elucidate the molecular link between the two conditions. Exendin-4, a peptide agonist of the glucagon-like peptide receptor recognized for its role in insulin secretion and as a versatile therapeutic agent for diabetes, [128] was administered to APP/PS1 transgenic mice along with infliximab. This intervention resulted in a notable reduction in hippocampal serine phosphorylation of insuline receptor substrate-1 (IRS-1), an event frequently observed in both diabetes and AD [129]. Aβ peptide oligomers activated the JNK/TNF-α pathway, leading to IRS-1 phosphorylation. Remarkably, treatment with exendin-4 demonstrated a decrease in hippocampal levels of serine IRS-1p levels and a concurrent improvement in cognitive function, as assessed through the Morris water maze test [125].

### 6.6. Atorvastatin

In addition, atorvastatin, typically prescribed as a cholesterol-lowering medication, has been utilized as an anti-TNF agent in AD Wistar rat models exposed to Aβ1-42 peptides. The group treated with atorvastatin exhibited a notable reduction in TNF-α levels. Furthermore, the assessment of neuroinflammatory cytokines, including IL-1β and IL-6 through immunostaining, revealed a decrease. This reduction contributed to enhanced learning and memory capabilities, along with the mitigation of neuronal damage [130]. For this reason, atorvastatin, used as an anti-TNF agent, represents a potential therapeutic drug that reduces TNF-α, neuroinflammatory cytokines, enhancing learning, memory, and mitigating neuronal damage in AD rat models.

### 6.7. Infliximab, Etanercept and Adalimumab

Shi et al. conducted an investigation into the therapeutic efficacy of infliximab administered via intracerebroventricular injection in APP/PS1 mice. The study elucidates the impact of infliximab on various neurobiological factors. Specifically, the administration of infliximab resulted in a notable reduction in TNF-α levels within the brain. This reduction correlated with diminished Aβ plaques and decreased phosphorylation of the pTau protein [131]. Intriguingly, the treatment exhibited an augmentative effect on CD11c-positive dendritic cells, which are thought to play a significant role in Aβ deposition [132]. In summary, infliximab intracerebroventricular administration reduces TNF-α, Aβ plaques, and pTau phosphorylation and potentially influences CD11c-positive dendritic cells in AD.

In a study by Kübra Elçioğlu et al., the impact of systemic thalidomide and intracerebroventricular administration of etanercept and infliximab was investigated in a rat model of intracerebroventricular streptozotocin-induced dementia. Rats were categorized into four groups, each receiving a distinct drug, including a control group. Memory tests, specifically the Morris water maze and passive avoidance tests, were conducted. The results indicated that all treatment groups exhibited preventive effects on learning and memory deficits. Notably, the streptozotocin-thalidomide group demonstrated significantly superior performance in the memory performance [133].

Adalimumab, explored as a potential anti-TNF-α monoclonal antibody in Aβ1-40 mice, was investigated by Park and colleagues [134]. Their study revealed that adalimumab treatment not only reduced neuronal damage but also mitigated neuroinflammation. This beneficial effect was attributed to the downregulation of BACE1 protein expression and the attenuation of Aβ1-40 plaques. Adalimumab was also found to increase the levels of brain-derived neurotrophic factor (BDNF) expression. Additionally, it demonstrated a regulatory impact on the TNF-α signaling pathway, as evidenced by the attenuation of phosphorylation at residues of NF-κB, which is typically heightened in this pathway [134]. In conclusion, adalimumab, an anti-TNF-α antibody, reduces neuronal damage, mitigates neuroinflammation, and regulates key proteins, offering potential therapeutic benefits in Aβ1-40 mice.

In a recent study by Li and colleagues [135,136,137], etanercept was administered via intravenous injection to 3xTg-AD transgenic mice. The authors found that etanercept significantly increased spatial memory, long-term memory, and working memory, as assessed using the Morris water maze and Y-maze tasks. Etanercept treatment also led to a reduction in neuronal injury and reduced cytokine levels in AD mice. Etanercept effectively inhibited the activation of c-Jun N-terminal Kinases (JNK) and NF-κB pathways in vivo, which represent crucial pathways in neuroinflammation [136,137]. Etanercept enhances spatial and long-term memory, reduces neuronal injury, and inhibits key neuroinflammatory pathways in 3xTg-AD mice.

### 6.8. TfRMAb-TNFR

In another recent study, Ou and colleagues investigated the therapeutic impact of TfRMAb-TNFR, a fusion protein designed to target TNF-α, in AD. This molecule was generated by an innovative approach merging the TNFR, a biological inhibitor of TNF-α (TNFI) that sequesters TNF-α and a transferrin receptor antibody (TfRMAb) to facilitate the delivery of the TNFI. The study yielded positive results when administering TfRMAb-TNFR to 10.7-month-old APP/PS1 mice, with a reduction in the levels of Aβ plaques, in the insoluble-oligomeric Aβ, and in the levels of brain microgliosis. Remarkably, the TfRMAb component enabled its penetration through the blood–brain barrier, showcasing its ability to access the CNS. The treatment with TfRMAb-TNFR was associated with an enhancement in the expression of tight junction proteins, contributing to its therapeutic efficacy [138]. In summary, TfRMAb-TNFR reduces Aβ plaques, Aβ oligomers, and microgliosis, and enhances tight junction proteins, demonstrating therapeutic potential in AD.

In conclusion, the entirety of these studies reveals a diverse range of TNF-α inhibitors in preclinical studies, as in animal studies promoting the notion of their future efficacy as potential pharmaceutical agents.

Table 2 summarizes the main evidence coming from animal studies.

## 7. Future Directions

Going through the evidence related to the application of TNF-α inhibition in AD highlights how this represents a very promising therapeutic approach. Therefore, future directions in this research need to clarify and optimize it. TNF-α inhibitors have a possible application in many aspects of AD pathogenesis, from microglial activation to the reduction of neuronal loss and the amelioration of blood–brain barrier integrity. Exploring these potential applications and optimizing the use of TNF-α inhibitors could pave the way for more effective and targeted treatments in the future.

In future studies, some key aspects will need clarification. The first of these is represented by the penetration of anti-TNF-α drugs through the blood–brain barrier, given the large molecular weight of anti-TNF-α monoclonal antibodies which makes the passage impossible under physiological conditions. However, the blood–brain barrier becomes increasingly permeable and damaged in AD and is characterized by a decrease in the expression of tight junction proteins [139]. The second aspect is the best timing for TNF-α inhibition in the disease course, as evidenced by some of the results of animal models (e.g., minocycline studies). In fact, early administration may have a different impact on the characteristic pathological features of AD as compared to late treatment. Finally, whether the coadministration of TNF-α inhibitors and the other drugs commonly prescribed in AD (including galantamine, rivastigmine, donepezil, memantine, and the recently approved monoclonal antibodies that bind to Aβ protofibrils, lecanemab, and aducanumab) is feasible is still unknown. The limited evidence currently available on this matter suggests that the combination of rivastigmine with leflunomide demonstrated a significant decrease in TNF-α levels [140]. Rivastigmine also exhibited anti-inflammatory effects in mice by reducing TNF-α and IL-6 release from macrophages [141]. The protective effects against TNF-α-induced damage of donepezil are still uncertain [142,143] and therefore its combination with TNF-α inhibitors warrants further investigation.

In the realm of AD treatment, the concepts of personalized medicine and pharmacogenomics have gained significance, and these aspects must be considered also in the context of TNF-α inhibition. Personalized medicine tailors medical interventions to individual characteristics, including genetic makeup, to optimize treatment efficacy. Pharmacogenomics explores how an individual’s genetic profile influences their response to drugs. Therefore, other aspects to consider for future therapeutic approaches will also include tailoring TNF-α inhibition strategies based on individual patient profiles and genetic factors and finally the evaluation of the long-term safety and efficacy of these drugs in a peculiar population such as AD patients. For instance, anti-TNF-α inhibition has been associated with acute infusion reactions (which include symptoms like headache, fever, chills, urticaria, chest pain), infections, tumor development and few cases of drug-induced lupus, seizures, and pancytopenia [144,145,146]. All of these may represent matters of concern for AD patients.

From this perspective, the availability of targeted genetic investigations, together with the study of the characteristic immune response of AD, could prove highly valuable.

## 8. Conclusions

The evidence collected so far on the role played by TNF-α in AD emphasizes that this molecule impacts various neuronal activities, including modulation of neurotransmission, sleep regulation, and astrocyte-mediated glutamate release. TNF-α is a key molecule in orchestrating chronic inflammation and can influence the synthesis of Aβ plaques, and the formation of neurofibrillary tangles and therefore can curb the progression of AD pathology. Clinical trials exploring the effects of TNF-α inhibitors in the development and progression of AD hold interesting results. Interesting data also came from the investigation of several molecules characterized by TNF-α inhibition in animal models of AD. These results show promising evidence to target disease progression by addressing different inflammatory pathways induced by TNF-α.

Unfortunately, we do not have enough data from clinical trials testing the efficacy of anti-TNF-α drugs in AD. All the available trials are in the very early phases, and current evidence is therefore absolutely limited. Future high-quality studies evaluating the therapeutic potential of anti-TNF-α drugs in AD are warranted and this line of research should be pursued in the near future.

## Figures and Tables

**Table 1 cells-13-00054-t001:** Clinical trials exploring the efficacy of anti-TNF-α drugs on the progression of Alzheimer’s disease.

Reference	Type of Study	Intervention	Main Findings
Tobinick et al., 2006 [107]	prospective, single-center, open-label, pilot study	25–50 mg of etanercept was administered once weekly by perispinal administration for 6 months	significant improvement in cognition in treated patients
Tobinick et al., 2008 [109]	prospective, single-center, open-label, pilot study	weekly administration of etanercept, 25–50 mg, perispinally for six months in 12 patients with mild to severe AD for 6 months	Significant improvement in cognition in treated patients and rapid improvement in verbal fluency and aphasia in two dementia patients, starting minutes after administration of perispinal etanercept.
Butchart et al., 2015 [106]	randomized, placebo-controlled, double-blind, phase 2 trial registered with EudraCT (2009-013400-31) and ClinicalTrials.gov (NCT01068353)	peripheral subcutaneous administration of etanercept 50 mg once weekly for 24 weeks; 41 participants with mild to moderate Alzheimer disease	Treatment well tolerated but no significant change in cognition, behavior, or global function.
Chen et al., 2010 [105]	pilot study	15 elderly patients with rheumatoid arthritis: 8 received etanercept 25 mg twice weekly and 7 received adalimumab 40 mg twice monthly	Cognitive improvement in 11 of 15 participants; no improvement in depression.

**Table 2 cells-13-00054-t002:** Main evidence coming from animal studies. 3,6-DT 3,6-Dithiothalidomide; Aβ amyloid beta; AD Alzheimer’s disease; APP amyloid precursor protein; ELISA Enzyme Linked Immunosorbent Assay; IL interleukin; NaHS hydrogen sulfide; PS1 presenilin 1; RT-PCR reverse transcription polymerase chain reaction; tg transgenic; TfRMAb transferrin receptor antibody; TNF-α tumor necrosis factor α; TNFR TNF-α tumor necrosis factor α receptor.

Reference	Animal Model	Compound	Methods	Main Findings
Seabrook et al., 2006 [114]	APP-tg mice	Minocycline	ELISA and memory and cognitive testing	Increasing in Aβ deposition but cognitive performance improved. Administration post-Aβ deposition suppressed microglial activation without impacting in Aβ levels or cognitive function.
Biscaro et al., 2012 [115]	APP/PS1 mice	Minocycline	ELISA, cognitive testing and memory and cognitive testing	minocycline treatment increased the survival of dentate granule cells
Chavant et al., 2010 [116]	APP-tg mice	Imipramine	Western blottings, ELISA, cognitive testing and memory and cognitive testing	reduce elevated Aβ levels
Shi et al., 2011 [131]	APP/PS1 mice	Infliximab	Immunohistochemistry	Reduction in TNF-α levels, Aβ plaques and decreased pTau protein. An increase in CD11c-positive dendritic cells.
Xuan et al., 2012 [118]	Wistar rats	NaHS	RT-PCR, analysis, memory and cognitive testing and working memory testing.	NaHS mitigates the presence of Aβ1-40 demonstrating a neuroprotective role and a reduction in the expression of TNF-α and IL-1 β levels.
Gabbita et al., 2012 [122]	3xTg-AD mice	Thalidomide and its analog 3,6-DT	Immunohistochemistry, RT-PCR analysis, and memory and cognitive testing.	An enhancement in working memory performance. 3,6-DT prevents cognitive impairment.
Belarbi et al., 2012 [123]	AD mice	3,6-DT	Immunohistochemistry, and cognitive testing.	The treatment successfully normalized TNF-α levels, although IL-1β remained elevated.
Tweedie et al., 2012 [124]	3xTg-AD mice	3,6-DT	RT-PCR, Immunohistochemistry, and cognitive testing.	Decreased levels of pTau protein, Aβ peptide, and Aβ plaque numbers and memory improvement.
Bomfim et al., 2012 [125]	APP/PS1 mice	Exendin-4 along with Infliximab	Immunohistochemistry, memory and cognitive testing.	Association between impaired brain insulin levels and cognitive impairments.
X.-H. Li et al., 2013 [130]	Wistar rats	Atorvastatin	Immunohistochemistry, Memory and working memory testing.	Assessment of decreasing in neuroinflammatory cytokines, including IL-1β, IL-6 and TNF-α levels.
Kübra Elçioğlu et al., 2015 [133]	Sprague–Dawley mice	Thalidomide, Etanercept and Infliximab	Memory and cognitive testing.	Results indicated that all treatment groups exhibited preventive effects on learning and memory deficits, in particular the thalidomide group.
Park et al., 2019 [134]	Aβ1-40 mice	Adalimumab	Immunohistochemistry, Memory and working memory testing.	Adalimumab treatment not only reduced neuronal damage but also mitigated neuroinflammation.
Y. Li et al., 2022 [135]	3xTg-AD	Etanercept	Immunohistochemistry, Memory and working memory testing.	Etanercept treatment significantly enhanced spatial memory, long-term memory, and working memory and lowered cytokine levels in AD mice.
Ou et al., 2022 [138]	APP/PS1 mice	TfRMAb-TNFR	Immunohistochemistry, Memory and working memory testing.	TfRMAb-TNFR exerted protective effects, effectively reducing the levels of Aβ plaques.

## Data Availability

No new data were created.

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
