# Peer review of "The Role of TNF-α in Alzheimer’s Disease: A Narrative Review"

_cells, 2023, doi:10.3390/cells13010054_

Round 1
Reviewer 1 Report
Comments and Suggestions for Authors
Interesting and up-to-date review. it is certainly a valuable addition to the literature. I have a few suggestions:
line 37 "These deposits disrupt..." Given that the Authors present alternative hypotheses and the exact pathogenic role of amyloid its still controversial (Kasper P Kepp, Nikolaos K Robakis, Poul F Høilund-Carlsen, Stefano L Sensi, Bryce Vissel, The amyloid cascade hypothesis: an updated critical review, Brain, Volume 146, Issue 10, October 2023, Pages 3969–3990, https://doi.org/10.1093/brain/awad159), I would recommend to rephrase as "these deposits supposedly disrupt"
line 79 maybe the Authors could mention how many articles or the percentage thereof that was rejected.
Author Response
Point 1: Interesting and up-to-date review. it is certainly a valuable addition to the literature. I have a few suggestions:
line 37 "These deposits disrupt..." Given that the Authors present alternative hypotheses and the exact pathogenic role of amyloid its still controversial (Kasper P Kepp, Nikolaos K Robakis, Poul F Høilund-Carlsen, Stefano L Sensi, Bryce Vissel, The amyloid cascade hypothesis: an updated critical review, Brain, Volume 146, Issue 10, October 2023, Pages 3969–3990, https://doi.org/10.1093/brain/awad159), I would recommend to rephrase as "these deposits supposedly disrupt"
Response 1: We thank reviewer #1 for her/his critiques. As suggested, we modified the sentence as follows: “These deposits supposedly disrupt normal synaptic communication and, in the end, may contribute to neurodegeneration.[4] The reference has therefore added to replace the previous one.”
Point 2: line 79 maybe the Authors could mention how many articles or the percentage thereof that was rejected.
Response 2: We thank reviewer #1 for her/his critiques. We added the requested details in the methods.

Reviewer 2 Report
Comments and Suggestions for Authors
This is a very well written and interesting review. However, I have some comments and suggestions, which may improve the quality of this paper:
Introduction
Here are some suggestions to improve this text:
- The introduction could be more concise. The first 3 paragraphs provide background on Alzheimer's disease but do not need so much detail before getting to the main focus on the role of TNF-alpha. I'd recommend shortening those paragraphs.
- There is some repetitiveness discussing inflammation and the immune system's role in paragraphs 4-5. Consider combining some of those points.
- The statement "These deposits disrupt normal synaptic communication..." in paragraph 3 seems out of place. I'd recommend moving it to paragraph 4 where amyloid plaques and neurofibrillary tangles are first mentioned.
- Paragraph 5 introduces a number of additional hypotheses about Alzheimer's pathogenesis. Unless the rest of the text will be covering those, they distract from the main focus on neuroinflammation. I'd suggest removing that sentence.
- Paragraphs 6-8 provide very helpful context about the key role TNF-alpha plays in neuroinflammation and potential negative effects in AD. Well done there.
- The last paragraph nicely summarizes the goals of the review. However, the text preceding this is supposed to be the actual review content. Consider framing those sections more clearly as reviewing existing evidence about the specific role and therapeutic implications of TNF-alpha in AD.
TNF-α metabolism
- Overall, the text provides a good overview of the complex roles TNF-alpha plays in the central nervous system and how dysregulation may contribute to Alzheimer's pathogenesis. However, it does jump around a bit between different mechanisms and effects, making it slightly hard to follow. Consider reorganizing some paragraphs to improve flow.
- The introductions to the various functions of TNF-alpha (paragraphs 5-8) are interesting but quite lengthy before getting to the main focus on Alzheimer's disease. As this is meant to be a focused review on AD, consider condensing background details to just key points needed to support that focus.
- Paragraph 4 provides rationale for TNF-alpha's involvement in AD, but the actual evidence is not discussed until paragraph 21. Consider moving some of that evidence earlier in the text to strengthen the rationale.
- There are some lengthy sentences with complex clauses (e.g. paragraph 16, paragraph 20). Simplifying sentence structures could improve clarity.
- The abbreviations for the TNF receptors (TNFR1, TNFR2) are introduced in paragraph 16 before the receptors themselves are explained. Define them at first use.
- Paragraph 19 mentions "TRAF2 recruitment" and "PI3K/NF-κB pathway" without explanation. Consider briefly clarifying specialized terms.
- Paragraph 21 states evidence in humans will be reviewed next, but the text ends abruptly without delivering on this. If this is an incomplete draft, make that clear.
Evidence in humans
- The introduction in paragraph 1 mentions various types of evidence that will be discussed, but the actual structure of the sections seems a bit disjointed. Consider organizing by type of evidence (CSF studies, genetic studies, etc.) for better flow.
- Several studies are well explained in paragraphs 2-4, but paragraphs 5-7 then jump between different study methods quite quickly. Slow this down a bit - it reads rushed.
- Abbreviations like MCI, CSF, MMSE should be defined at first use for clarity.
- Paragraph 8 mentions interesting results but does not describe the study methods. Add details on the populations, techniques, etc. used.
- Paragraph 9 brings up promising findings on TNF inhibitor therapies, but mainly focuses on rheumatoid arthritis. Since the focus is Alzheimer's, consider shortening details about other conditions.
- The summarized studies in paragraphs 15-16 lack sufficient context and details from the original research. Expand on the key methods, patient groups, interventions, and findings.
- The conclusion in paragraph 17 refers to "completed clinical trials" but trials were never explicitly discussed, just various studies. Rephrase this to accurately reflect content.
- Addressing writing style - some long, complex sentences could be simplified to improve readability.
Evidence in animal models of Alzheimer’s disease
- The introduction mentions various TNF-alpha inhibitors that have been tested, but the flow into summarizing specific studies feels abrupt. Consider a sentence or two to transition and provide context on the studies that will be reviewed.
- The summaries of each animal study are clear, but adding a sentence on the key takeaway or significance of findings for each study would make the importance of each one clearer to the reader.
- The sequence of study summaries seems a bit haphazard. Consider reorganizing by type of intervention (minocycline studies, thalidomide studies, etc.) for better flow.
- Details on the methodologies and models used are currently lacking. Add a sentence or two on study details - animal models, how AD was induced, numbers, etc. - to provide more context.
- The text ends quite abruptly without any concluding thoughts. Add a closing paragraph highlighting the key takeaways and significance of these preclinical findings on targeting TNF-alpha for AD.
Future Directions
- The opening sentence summarizes the evidence well. However, the next sentence about future directions feels abrupt. Consider starting a new paragraph here to mark the transition.
- When mentioning future research questions in paragraph 2, be more specific on the key gaps the field needs to address. For example, which aspects of BBB penetration need clarification? What is the current evidence on timing of treatment?
- The mention of co-administration with other AD drugs in paragraph 2 is interesting but vague. Are there any theoretical issues? Evidence so far? Elaborate.
- Paragraph 3 brings up personalized medicine and pharmacogenomics but this comes out of nowhere. Introduce these ideas earlier so the conclusion ties together previous threads.
- The endpoints in paragraph 3 around safety and efficacy evaluations are crucial but could apply to any drug. Tailor these to issues more specific to anti-TNF therapies.
- There are multiple complex sentences, especially paragraphs 2-3, that could be simplified for clarity and easier reading comprehension.
Author Response
Reviewer #2
This is a very well written and interesting review. However, I have some comments and suggestions, which may improve the quality of this paper:
Introduction
Here are some suggestions to improve this text:
Point 1: The introduction could be more concise. The first 3 paragraphs provide background on Alzheimer's disease but do not need so much detail before getting to the main focus on the role of TNF-alpha. I'd recommend shortening those paragraphs.
Response 1: As suggested by reviewer #2, we condensed the first 3 paragraphs by synthesizing them into a single introductory paragraph.
Therefore the text changed from this:
“Alzheimer's disease (AD) is a progressive neurodegenerative disorder that represents the most common cause of dementia in the elderly.[1] Its incidence and prevalence are increasing with the aging of the population, posing a growing public health concern. Typical clinical manifestations include declarative memory deterioration, temporal and spatial disorientation, language difficulties, judgment problems, and other cognitive impairments.[2]
Known risk factors for AD include advanced age, genetic predisposition, diabetes, hypertension, obesity, smoking, lack of physical activity, and low educational attainment.[3]
The key pathological feature of AD is the accumulation of beta-amyloid (Aβ) plaques and neurofibrillary tangles.[4] Aβ plaques consist of aggregates of Aβ protein, while neurofibrillary tangles are composed of hyperphosphorylated tau protein filaments. These deposits disrupt normal synaptic communication and, in the end, cause neurodegeneration. However this “classical” pathogenic model does not explain much of the clinical and pathological variability of the disease, therefore several additional hypotheses have been formulated.[5] These include the tau propagation hypothesis,[6,7] mitochondrial dysfunction,[8–11] calcium homeostasis alteration,[12] neurovascular hypothesis,[13] metal ion hypothesis,[14] and lymphatic system hypothesis.[15] The "inflammatory hypothesis" of AD [16,17] suggests that chronic inflammation in the brain plays a significant role in the development and progression of the condition. Ongoing immune system activation and the release of pro-inflammatory molecules contribute to the characteristic AD pathological changes, with the final result of damaging neurons, and im-pairing the clearance of toxic substances. Recent studies highlight that inflammation is a central feature in the complex pathophysiology of AD.[18–21] The intricate interplay between the central nervous system (CNS) and the peripheral immune system has come into focus, revealing the multifaceted contributions of various immune cell types and molecules, both in the peripheral blood and in the CNS.[22,23] Altered levels of several cytokines have been observed in the peripheral blood and cerebrospinal fluid (CSF) of patients with AD.[24–27] TNF-α, a proinflammatory cytokine, has attracted attention due to its multifaceted and still not fully characterised role in AD and the fact that it could potentially represent a future therapeutic target. Indeed, TNF-α has many roles, not only in inflammation, but also in cell death and proliferation, and is crucial in maintaining CNS homeostasis.[28–30] TNF-α is primarily produced by microglia, astrocytes, and neurons in response to various stimuli.[31] While TNF-α has a fundamental role in modulating excitotoxicity, neuroinflammation, blood-brain barrier permeability,[29] regulation of oligodendrocyte survival, myelin formation and repair,[32] its excessive or dysregulated production can have detrimental effects on neuronal health, contributing to the development of neurodegeneration. [33]”
To this:
“Alzheimer's disease (AD) is a progressive neurodegenerative disorder that represents the most common cause of dementia in the elderly.[1] Typical clinical manifestations include declarative memory deterioration, temporal and spatial disorientation, language difficulties, judgment problems, and other cognitive impairments [2]. The key pathological feature of AD is the accumulation of beta-amyloid (Aβ) plaques and neurofibrillary tangles.[3] Aβ deposits supposedly disrupt normal synaptic communication and, in the end, may contribute to neurodegeneration.[4] However this “classical” pathogenic model does not explain much of the clinical and pathological variability of the disease, therefore several additional hypotheses have been formulated.[5] The "inflammatory hypothesis" of AD [6,7] suggests that chronic inflammation in the brain plays a significant role in the development and progression of the condition. Ongoing immune system activation and the release of pro-inflammatory molecules contribute to the characteristic AD pathological changes, with the final result of damaging neurons, and impairing the clearance of toxic substances.[18–21] The intricate interplay between the central nervous system (CNS) and the peripheral immune system has come into focus, [12,13] revealing altered levels of several cytokines in the peripheral blood and cerebrospinal fluid (CSF) of patients with AD.[14–17] Tumor necrosis factor (TNF)-α, a proinflammatory cytokine, has attracted attention due to its multifaceted and still not fully characterised role in AD and the fact that it could potentially represent a future therapeutic target. Indeed, TNF-α has many roles, not only in inflammation, but also in cell death and proliferation, and is crucial in maintaining CNS homeostasis.[18–20] TNF-α is primarily produced by microglia, astrocytes, and neurons in response to various stimuli.[21] While TNF-α has a fundamental role in modulating excitotoxicity, neuroinflammation, blood-brain barrier permeability,[19] regulation of oligodendrocyte survival, myelin formation and repair,[22] its excessive or dysregulated production can have detrimental effects on neuronal health, contributing to the development of neurodegeneration. [23]”
Point 2: There is some repetitiveness discussing inflammation and the immune system's role in paragraphs 4-5. Consider combining some of those points.
Response 2: We tried to reduce the length and improve the repetitiveness of the text discussing inflammation (see previous modifications).
Point 3: The statement "These deposits disrupt normal synaptic communication..." in paragraph 3 seems out of place. I'd recommend moving it to paragraph 4 where amyloid plaques and neurofibrillary tangles are first mentioned.
Response 3: As already discussed, we condensed the first paragraph and, as also suggested by reviewer #1, we modified the text as follows: “The key pathological feature of AD is the accumulation of beta-amyloid (Aβ) plaques and neurofibrillary tangles.[3] Aβ deposits supposedly disrupt normal synaptic communi-cation and, in the end, may contribute to neurodegeneration.[4]
Point 4: Paragraph 5 introduces a number of additional hypotheses about Alzheimer's pathogenesis. Unless the rest of the text will be covering those, they distract from the main focus on neuroinflammation. I'd suggest removing that sentence.
Response 4: As suggested, the following sentence has been removed: “These include the tau propagation hypothesis,[6,7] mitochondrial dysfunction,[8–11] calcium homeostasis alteration,[12] neurovascular hypothesis,[13] metal ion hypothesis,[14] and lymphatic system hypothesis.[15]”
Point 5: Paragraphs 6-8 provide very helpful context about the key role TNF-alpha plays in neuroinflammation and potential negative effects in AD. Well done there.
Response 5: We thank the reviewer for her/his comment.
Point 6: The last paragraph nicely summarizes the goals of the review. However, the text preceding this is supposed to be the actual review content. Consider framing those sections more clearly as reviewing existing evidence about the specific role and therapeutic implications of TNF-alpha in AD.
Response 6: We tried to explain the sections more clearly, as suggested. The text has been modified from this:
“In this narrative review, we explored the role of TNF-α in the pathogenesis of AD. We first discussed the molecular structure, receptors, metabolism of TNF-α, and the putative mechanisms by which this cytokine is thought to modulate neuroinflammation in AD. We also delved into both preclinical and clinical studies providing valuable cues for the role of TNF-α in AD. We ultimately dealt with the potential therapeutic implications of targeting TNF-α in the course of the disease, to slow down the progression of cognitive function and possibly even prevent AD onset.[34–36]”
To this: “In this narrative review, we explored the role of TNF-α in the pathogenesis of AD. There are multiple pieces of evidence regarding the role of TNF-α, including its ability to induce amyloid precursor protein (APP) and β-site APP cleaving enzyme (BACE1) ex-pression in mouse astrocytes[24–26], its stimulation of Aβ synthesis, neuronal loss, and impairement microglia Aβ phagocytosis.[27]
We first discussed the molecular structure, receptors, and metabolism of TNF-α. We explored TNF-α molecular structure, with reference to its membrane-bound form and its soluble form, the two TNF-α receptors, and the metabolism of this cytokine. Then, we explained the rationale for the TNF-α involvement in AD. In this part, we summarize the activities in which TNF-α is involved within the CNS, the roles of TNF-α receptors in CNS physiology, and how TNF-α influences and curbs neuroinflammation in AD. We also delved into both preclinical and clinical studies providing valuable cues for the role of TNF-α in AD. Therefore, we summarised the evidence in humans, also discussing the clinical trials exploring the efficacy of anti-TNF-α drugs on the progression of AD; and the main evidence coming from animal studies in AD models. We ultimately dealt with the future directions, with a reference to the possible applications of TNF-α inhibition to slow down the progression of cognitive function and possibly even prevent AD onset.[28–30]”
Point 6: TNF-α metabolism. Overall, the text provides a good overview of the complex roles TNF-alpha plays in the central nervous system and how dysregulation may contribute to Alzheimer's pathogenesis. However, it does jump around a bit between different mechanisms and effects, making it slightly hard to follow. Consider reorganizing some paragraphs to improve flow.
Response 6: We tried to apply the suggested modifications (see following responses).
Point 7: The introductions to the various functions of TNF-alpha (paragraphs 5-8) are interesting but quite lengthy before getting to the main focus on Alzheimer's disease. As this is meant to be a focused review on AD, consider condensing background details to just key points needed to support that focus.
Response 7: We thank reviewer #2 for this suggestion. We significantly simplified the paragraph entitled “3.1 Molecular structure” from this:
“The term “tumor necrosis factor” was first coined by O’Malley and colleagues in 1962 and this molecule was initially recognized for its role in initiating apoptosis.[37] The TNF-α gene is situated in the class III region of the major histocompatibility complex on chromosome 6, found between the HLA-B and HLA-DR genes.[38] TNF is present in two distinct forms: a membrane-bound form and a soluble form.[39] In the process of its synthesis, TNF-α initially stays within the double layer of the cell membrane. Subsequently, a specific enzyme, known as the TNF-converting enzyme (TACE/ADAM17/CD156q),[39] liberates the membrane-bound TNF-α, which is originally 26 kDa in size (233 amino acid residues), into the extracellular environment, where it becomes a soluble 17 kDa protein, through a phenomenon called “shedding”.[40] Membrane-bound TNF-α is cleaved by TACE between residues of alanine76 and valine77.[41] Soluble TNF-α forms a homotrimer comprising 17-kDa cleaved monomers, while transmembrane TNF-α also exists as a homotrimer with 26-kDa uncleaved monomers.[42] Membrane-bound and soluble forms of TNF-α are both biologically active, but their specific functions vary based on their respective locations.[43]”
To this: “The term “tumor necrosis factor” was first coined by O’Malley and colleagues in 1962 and this molecule was initially recognized for its role in initiating apoptosis.[31] The TNF-α gene is situated in the class III region of the major histocompatibility complex on chro-mosome 6, found between the HLA-B and HLA-DR genes.[32] TNF is present in two distinct forms: a membrane-bound form and a soluble form.[33] In the process of its synthesis, TNF-α initially stays within the double layer of the cell membrane. Subse-quently, a specific enzyme, known as the TNF-converting enzyme (TACE/ADAM17/CD156q),[33] liberates the membrane-bound TNF-α, which is origi-nally 26 kDa in size (233 amino acid residues), into the extracellular environment, where it becomes a soluble 17 kDa protein, through a phenomenon called “shedding”.[34] Membrane-bound TNF-α is cleaved by TACE between residues of alanine76 and va-line77.[35] Soluble TNF-α forms a homotrimer comprising 17-kDa cleaved monomers, while transmembrane TNF-α also exists as a homotrimer with 26-kDa uncleaved monomers.[36] Membrane-bound and soluble forms of TNF-α are both biologically active, but their specific functions vary based on their respective locations.[37]”
Point 8: Paragraph 4 provides rationale for TNF-alpha's involvement in AD, but the actual evidence is not discussed until paragraph 21. Consider moving some of that evidence earlier in the text to strengthen the rationale.
Response 8: As already commented, we changed the following part from this:
“In this narrative review, we explored the role of TNF-α in the pathogenesis of AD. We first discussed the molecular structure, receptors, metabolism of TNF-α, and the putative mechanisms by which this cytokine is thought to modulate neuroinflammation in AD. We also delved into both preclinical and clinical studies providing valuable cues for the role of TNF-α in AD. We ultimately dealt with the potential therapeutic implications of targeting TNF-α in the course of the disease, to slow down the progression of cognitive function and possibly even prevent AD onset.[34–36]”
To this:
“In this narrative review, we explored the role of TNF-α in the pathogenesis of AD. There are multiple pieces of evidence regarding the role of TNF-α, including its ability to induce amyloid precursor protein (APP) and β-site APP cleaving enzyme (BACE1) ex-pression in mouse astrocytes[24–26], its stimulation of Aβ synthesis, neuronal loss, and impairement microglia Aβ phagocytosis.[27]
We first discussed the molecular structure, receptors, and metabolism of TNF-α. We explored TNF-α molecular structure, with reference to its membrane-bound form and its soluble form, the two TNF-α receptors, and the metabolism of this cytokine. Then, we explained the rationale for the TNF-α involvement in AD. In this part, we summarize the activities in which TNF-α is involved within the CNS, the roles of TNF-α receptors in CNS physiology, and how TNF-α influences and curbs neuroinflammation in AD. We also delved into both preclinical and clinical studies providing valuable cues for the role of TNF-α in AD. Therefore, we summarised the evidence in humans, also discussing the clinical trials exploring the efficacy of anti-TNF-α drugs on the progression of AD; and the main evidence coming from animal studies in AD models. We ultimately dealt with the future directions, with a reference to the possible applications of TNF-α inhibition to slow down the progression of cognitive function and possibly even prevent AD onset.[28–30]”
Point 9: There are some lengthy sentences with complex clauses (e.g. paragraph 16, paragraph 20). Simplifying sentence structures could improve clarity.
Response 9: We thank reviewer #2 and we simplified some sentences, e.g.:
- “Membrane-bound TNF-α Transmembrane TNF-α exhibits interactions with both TNF receptor 1 (TNFR1) and TNF receptor 2 (TNFR2); however, its primary mediation of biological activities is believed to occur through TNFR2. Soluble TNF-α functions in an endocrine and paracrine manner, as it can travel through the bloodstream to distant sites and affect cells not in direct contact with its source” was modified as follows: “Membrane-bound TNF-α exhibits a primary interaction with TNF receptor 2 (TNFR2) but can also bind TNF receptor 1 (TNFR1).Soluble TNF-α functions in an endocrine and paracrine manner, as it can travel through the bloodstream to distant sites.”
- “This rapid turnover enables TNF-α to respond promptly to alterations in the microenvironment, contributing to the finely tuned regulation of immune and inflammatory responses.” Was modified as follows: “This rapid turnover enables TNF-α to respond promptly to alterations in the microenvironment.”
- “Following the release, TNF-α undergoes metabolism and clearance from the bloodstream, largely orchestrated by the liver and kidneys.” Was modified as follows: “Following the release, TNF-α undergoes liver and kidney metabolism and clearance.”
Point 10: The abbreviations for the TNF receptors (TNFR1, TNFR2) are introduced in paragraph 16 before the receptors themselves are explained. Define them at first use.
Response 10: We added the following sentence at the beginning of the paragraph in order to define TNFR1 and TNFR2 at first use in the text: “There are two receptors for TNF-α, named TNF receptor 1 (TNFR1) and TNF receptor 2 (TNFR2).”
Point 11: Paragraph 19 mentions "TRAF2 recruitment" and "PI3K/NF-κB pathway" without explanation. Consider briefly clarifying specialized terms.
Response 11: The following sentences were added:
- TRAF family proteins are centrally involved in modulating inflammation, orchestrating antiviral defenses, and regulating apoptosis.
- The PI3K-Akt pathway serves as an intracellular signaling mechanism, driving essential cellular processes such as metabolism, proliferation, cell survival, growth, and angiogenesis in response to external signals. This regulatory cascade involves the phosphorylation of downstream substrates, particularly on serine and/or threonine residues.
Point 12: Paragraph 21 states evidence in humans will be reviewed next, but the text ends abruptly without delivering on this. If this is an incomplete draft, make that clear.
Response 12: We believe the critique stems from reviewer #2 encountering text display issues. Importantly, our version doesn't have sudden interruptions. In fact, the paragraph preceding the section on human evidence reads as follows: “Altogether, these insights underscore the intricate interplay of immune responses in AD pathogenesis and highlight the role of TNF-α. In the following paragraph, we will review the main evidence in human and animal studies and finally, we will discuss the possible future directions.”
Point 13:Evidence in humans The introduction in paragraph 1 mentions various types of evidence that will be discussed, but the actual structure of the sections seems a bit disjointed. Consider organizing by type of evidence (CSF studies, genetic studies, etc.) for better flow.
Response 13: We organized the text in subparagraphs entitled “5.1 CSF and blood studies”; “5.2 Genetic studies” and “5.3 TNF-α inhibitors”
Point 14: Several studies are well explained in paragraphs 2-4, but paragraphs 5-7 then jump between different study methods quite quickly. Slow this down a bit - it reads rushed.
Response 14: We agree with reviewer #2. The following sentences have been rephrased:
From: “Furthermore, another study outlined swift cognitive enhancement, starting within minutes, through the use of the same anti-TNF treatment approach in a late-onset AD patient. The rapid cognitive improvement observed with perispinal etanercept has been hypothesized to be linked to the mitigation of the effects of excess TNF-α on gliotransmission or other synaptic mechanisms in AD.[119,120]”
To: “Furthermore, another case report describing an 81-year-old patient who was given 25 mg of perispinal etanercept by posterior cervical interspinous injection outlined swift cognitive enhancement, starting within minutes. The rapid cognitive improvement observed in this patient has been hypothesized to be linked to the mitigation of the effects of excess TNF-α on gliotransmission or other synaptic mechanisms in AD.”
Point 14: Abbreviations like MCI, CSF, MMSE should be defined at first use for clarity.
Response 14: As suggested we defined MCI. CSF and MMSE were already defined.
Point 15: Paragraph 8 mentions interesting results but does not describe the study methods. Add details on the populations, techniques, etc. used.
Response 15: We agree with reviewer #2 for her/his suggestions and the text was modified as follows:
“Moreover, extensive research has suggested that specific TNF-α gene polymorphisms contribute to an increased risk of AD. In a meta-analysis aiming to define the association of common TNF-α gene polymorphisms with the risk of AD, Di Bona et al. selected 17 studies and evaluated them with a model-free method approach, to comprehensively analyze the results of these case-control genetic association studies. Notably, their research indicates a correlation between the −850 C > T polymorphism and the susceptibility to AD. Additionally, Yang et al. performed a case-control study in the Southern China population, involving the use of Polymerase Chain Reaction-Sequence Specific Primers (PCR-SSP) to assess TNF-α genotypes and alleles in 112 sporadic AD patients and 121 controls. Additionally, they quantified serum TNF-alpha levels through radioimmunoassay. They found significantly higher levels of serum TNF-α in sporadic AD patients, and that both -308 A/G polymorphism and elevation of serum level of TNF-α were both associated with increased risk of AD.”
Point 16: Paragraph 9 brings up promising findings on TNF inhibitor therapies, but mainly focuses on rheumatoid arthritis. Since the focus is Alzheimer's, consider shortening details about other conditions.
Response 16: We thank reviewer #2 for her/his suggestion. The text was modified as follows:
“Observational investigations conducted on individuals with systemic inflammatory disorders treated with TNF-α inhibitors offer additional substantiation for the implication of TNF-α in the pathogenesis of AD. A large retrospective case-control analysis, encompassing 56 million adult patients afflicted with inflammatory conditions, revealed a diminished risk of AD development with the administration of TNF-α inhibitors. A previous nested case-control study revealed a higher AD prevalence among RA patients compared to those without rheumatoid arthritis, with a further significant increase in risk in those who were affected by chronic conditions including coronary artery disease, diabetes, and peripheral vascular disease. Notably, exposure to anti-TNF agents, particularly etanercept, was associated with a reduced risk of AD in these patients. Elderly rheumatoid arthritis patients showed improved cognitive performance with subcutaneous anti-TNF-α therapy using drugs like etanercept and adalimumab. In contrast, a double-blind study in mild and moderate AD patients treated with subcutaneous Etanercept did not show significant changes in cognitive function, behavior, and global functions though there was a positive trend in the anti-TNF-α treatment group. The limited effectiveness of subcutaneous anti-TNF-α therapies in AD patients may be attributed to the large molecular weight of anti-TNF-α monoclonal antibodies because this makes the passage through the blood-brain barrier impossible under physiological conditions.”
Point 17: The summarized studies in paragraphs 15-16 lack sufficient context and details from the original research. Expand on the key methods, patient groups, interventions, and findings.
Response 17: The required details are now specified: “Furthermore, another case report describing an 81-year-old patient who was given 25 mg of perispinal etanercept by posterior cervical interspinous injection outlined swift cognitive enhancement, starting within minutes. The rapid cognitive improvement observed in this patient has been hypothesized to be linked to the mitigation of the effects of excess TNF-α on gliotransmission or other synaptic mechanisms in AD.”
Point 18: The conclusion in paragraph 17 refers to "completed clinical trials" but trials were never explicitly discussed, just various studies. Rephrase this to accurately reflect content.
Response 18: We rephrased the sentence as follows: “Studies exploring the efficacy of anti-TNF-α drugs on the progression of AD are summarized in Table 1”
Point 19: Addressing writing style - some long, complex sentences could be simplified to improve readability.
Response 19: We thank the reviewer. As already discussed, some long and complex sentences have been simplified.
Point 20: Evidence in animal models of Alzheimer’s disease
The introduction mentions various TNF-alpha inhibitors that have been tested, but the flow into summarizing specific studies feels abrupt. Consider a sentence or two to transition and provide context on the studies that will be reviewed.
Response 20: We agree with the reviewer. The following sentence has therefore been deleted: “The diverse range of compounds tested reflects the ongoing efforts to find effective interventions for AD by targeting the inflammatory pathways associated with TNF-α.” And the following sentences have been added: “This diverse range of compounds, evaluated in animal models, showcases the multifaceted approaches undertaken in the pursuit of finding effective interventions for AD by targeting the inflammatory pathways associated with TNF-α. In the upcoming sections, we will review and analyze key studies on animal models that contribute to our understanding of the therapeutic potential of TNF-α inhibition for AD.”
Point 21: The summaries of each animal study are clear, but adding a sentence on the key takeaway or significance of findings for each study would make the importance of each one clearer to the reader.
Response 21: We thank the reviewer and we added the key takeaway or significance of findings for each compound, where it was not already present.
Point 22: The sequence of study summaries seems a bit haphazard. Consider reorganizing by type of intervention (minocycline studies, thalidomide studies, etc.) for better flow.
Response 22: We reorganized the paragraphs as suggested.
Point 23: Details on the methodologies and models used are currently lacking. Add a sentence or two on study details - animal models, how AD was induced, numbers, etc. - to provide more context.
Response 23: As suggested we added extra details in the text (e.g. better specified the rat models). For all the details the reader can refer to the table.
Point 24: The text ends quite abruptly without any concluding thoughts. Add a closing paragraph highlighting the key takeaways and significance of these preclinical findings on targeting TNF-alpha for AD.
Response 24: The following text was added: In conclusion, the entirety of these studies reveals a diverse range of TNF-α inhibitors in preclinical studies, as in animal studies promoting the notion of their future efficacy as potential pharmaceutical agents.
Point 25: Future Directions
The opening sentence summarizes the evidence well. However, the next sentence about future directions feels abrupt. Consider starting a new paragraph here to mark the transition.
Response 25: We agree with the reviewer. The beginning of the paragraph has been reformulated as follows: “Going through the evidence related to the application of TNF-α inhibition in AD highlights how this represents a very promising therapeutic approach. Therefore, future directions in this research need to clarify and optimize it. TNF-α inhibitors have a possible application in many aspects of AD pathogenesis, from microglial activation to the reduction of neuronal loss and the amelioration of blood-brain barrier integrity. Exploring these potential applications and optimizing the use of TNF-α inhibitors could pave the way for more effective and targeted treatments in the future.”
Point 26: When mentioning future research questions in paragraph 2, be more specific on the key gaps the field needs to address. For example, which aspects of BBB penetration need clarification? What is the current evidence on timing of treatment?
Response 26: We agree with the reviewer. The text has been modified as follows: “In future studies, some key aspects will need clarification. The first of these is represented by the penetration of anti- TNF-α drugs through the blood-brain barrier, given the large molecular weight of anti-TNF-α monoclonal antibodies which makes the passage impossible under physiological conditions. However, the blood-brain barrier becomes increasingly permeable and damaged in AD and is characterized by a decrease in the expression of tight junction proteins. The second aspect is the best timing for TNF-α inhibition in the disease course, as evidenced by some of the results of animal models (e.g. minocycline studies). In fact, early administration may have a different impact on the characteristic pathological features of AD as compared to late treatment.”
Point 27: The mention of co-administration with other AD drugs in paragraph 2 is interesting but vague. Are there any theoretical issues? Evidence so far? Elaborate.
Response 27: We thank reviewer #2 for her/his suggestion and we therefore modified the text as follows:
“Finally, whether the coadministration of TNF-α inhibitors and the other drugs commonly prescribed in AD (including galantamine, rivastigmine, donepezil, memantine, and the recently approved monoclonal antibodies that bind to Aβ protofibrils, lecanemab, and aducanumab) is feasible is still unknown. The limited evidence currently available on this matter suggests that the combination of rivastigmine with leflunomide demonstrated a significant decrease in TNF-α levels. [ref] Rivastigmine also exhibited anti-inflammatory effects in mice by reducing TNF-α and IL-6 release from macrophages. [ref] The protective effects against TNF-α-induced damage of donepezil are still uncertain [reff] and therefore its combination with TNF-α inhibitors warrants further investigation.”
Point 28: Paragraph 3 brings up personalized medicine and pharmacogenomics but this comes out of nowhere. Introduce these ideas earlier so the conclusion ties together previous threads.
Response 28: We thank the reviewer and we introduced the concepts by adding the following text: “In the realm of AD treatment, the concepts of personalized medicine and pharmacogenomics have gained significance, and these aspects must be considered also in the context of TNF-α inhibition. Personalized medicine tailors medical interventions based on individual characteristics, including genetic makeup, to optimize treatment efficacy. Pharmacogenomics explores how an individual genetic profile influences their response to drugs.”
Point 29: The endpoints in paragraph 3 around safety and efficacy evaluations are crucial but could apply to any drug. Tailor these to issues more specific to anti-TNF therapies.
Response 29: We thank the reviewer and we modified the text as follows: “For instance, anti-TNF-α inhibition has been associated with acute infusion reactions (which include symptoms like headache, fever, chills, urticaria, chest pain), infections, tumor development and few cases of drug-induced lupus, seizures, and pancytopenia.[reff] All these may represent matters of concern for AD patients.” Obviously, none can be said about the long-term efficacy of anti-TNF-α inhibition in AD.
Point 30: There are multiple complex sentences, especially paragraphs 2-3, that could be simplified for clarity and easier reading comprehension.
Response 30: We widely revised the text, as already commented, also taking into account these last suggestions.

Round 2
Reviewer 2 Report
Comments and Suggestions for Authors
The authors responded to my comments and suggestions very well. Thank you.